# Effects of Chelating Agents Addition on Ryegrass Extraction of Cadmium and Lead in Artificially Contaminated Soil

**Wen Dong** [1],*, **Ruichen Wang** [1], **Huaien Li** [1], **Xiao Yang** [2], **Jiake Li** [1], **Hui Wang** [1], **Chunbo Jiang** [1] and **Zhe Wang** [1]

[1] State Key Laboratory of Eco-Hydraulics in Northwest Arid Region of China, Xi'an University of Technology, Xi'an 710048, China; wangruichen@sina.com (R.W.); lhuaien@mail.xaut.edu.cn (H.L.); sys@xaut.edu.cn (J.L.); wanghui306@xaut.edu.cn (H.W.); chunbo@xaut.edu.cn (C.J.); wangzhewater@xaut.edu.cn (Z.W.)

[2] Institute of Geographical Sciences and Natural Resources Research, Chinese Academy of Sciences, Beijing 100101, China; yangx@igsnrr.ac.cn

\* Correspondence: dongwen@xaut.edu.cn

**Abstract:** This study investigated the removal of cadmium (Cd) and lead (Pb) from the soil through phytoremediation using ryegrass combined with chelating agents. Soil leaching experiments were employed to determine the extraction efficiencies of chelating agents, including ethylenediaminetetraacetic acid (EDTA), citric acid (CA), sodium glutamate tetra acetate (GLDA), oxalic acid (OA), and diethylenetriaminepentaacetic acid (DTPA) on Cd and Pb. Soil pot experiments were conducted to determine the effects of five different chelating agents—GLDA, EDTA, DTPA, CA, and OA—on the growth of ryegrass and the enrichment of Cd and Pb. The main findings were as follows: (1) the extraction efficiencies for Cd and Pb in soil were found to be GLDA > EDTA > DTPA > CA > OA and EDTA > DTPA > GLDA > CA > OA, respectively. (2) The aminopolycarboxylic acid class of chelating agents significantly reduced Cd and Pb contents in the weak acid extractable and reducible states in the studied soil, yet were less effective in the extraction of their residue state. Using chelating agents increased the proportion of residual heavy metals while reducing those in the weak acid extractable and reducible states in the soil, thereby mitigating the harmful effects of these heavy metals on the soil ecology.

**Keywords:** phytoremediation; cadmium; lead; chelating agents; ryegrass extraction

## 1. Introduction

Due to ongoing rapid economic development and expanding urbanization, increasingly large amounts of waste are being discharged into the soil [1,2]. Among these wastes, heavy-metal pollutants are particularly serious [3–7]. Heavy metals have high bioaccumulation, are recalcitrant and toxic, and their excessive accumulation in soil destroys its structure and biochemical processes [8,9]. Unlike organic contaminants, heavy metals do not undergo microbial or chemical degradation, and their total concentrations in soil persist for a long time [10], although some can change valencies. Heavy metals reach the soil environment through both pedogenic and anthropogenic processes, and the residence time for heavy metals in the topsoil (5 cm) can be decades, while at a depth of 20 cm, they can persist for centuries [11–15]. Heavy metals are a major concern for the environment and public health due to their toxicity. Cd is one of the most toxic elements to human beings, as it is harmful to bones and kidneys when accumulated over a long period of time. Pb is one of the most abundant toxic metals that exerts the most dangerous effects on all living beings. It has high mobility and is easily absorbed by plants, even at low contamination levels. The health hazards caused by Cd and Pb pollution have a time lag, meaning they take a long time to accumulate before they become apparent. The World Health Organization has classified Cd as a priority food contaminant for research [16–18]. According to the "National Survey Bulletin on Soil Pollution (China)" in 2014, the total exceeding the standard rate of soil pollution points reached 16.1%, and the over-standard

rate of Cd and Pb reached 7.0% and 1.5%, respectively; Cd had the largest proportion among all heavy metal pollutants, and Pb was the fifth among all heavy metals [19,20]. Given this, Cd and Pb were selected as the heavy metals to study in this paper.

Several methods have been developed and applied to remediate heavy metals, such as soil washing, solidification and stabilization, vitrification, electrokinetic remediation, in situ flushing, permeable reactive barriers, and monitored natural attenuation [21–26]. However, these methods are less feasible on a large scale because they are environmentally disruptive and cost-prohibitive [27]. Phytoremediation is a cost-effective remediation solution for removing pollutants, especially heavy metals, from contaminated soils at the site level with little disturbance to the landscape and can achieve the aims of green and sustainable remediation [28–30].

Phytoremediation is a mechanism for addressing heavy metal pollution that capitalizes on certain plants' ability to tolerate, hyperaccumulate, and decompose heavy metals. The process involves screening for plants that have high bioaccumulation potential and using them to absorb, transform, fix, and volatilize toxic heavy metals in contaminated soil through plants and microbial systems in the rhizosphere, thereby reducing heavy metal content in polluted soil through plant harvesting and recovery for remediation purposes. This approach is known as green remediation or bioremediation [31,32].

However, the natural phytoextraction process is limited by poor heavy metal bioavailability within the rhizosphere, which depends on soil pH and clay content, cellular tolerance to metals, soil nutrient levels, and metal selectivity [33,34]. To address these limitations, researchers have turned to chemically assisted phytoextraction, also referred to as chelate-enhanced phytoextraction. This technique involves adding chelating agents to improve the remediation efficiency of plants. Ethylenediaminedisuccinic acid (EDDS) and ethylenediaminetetraacetic acid (EDTA) are commonly used chelating agents [35–37]. Chelating agents increase the solubility of heavy metals, promoting their uptake by plants to remediate heavy metal-contaminated soil [38]. Although EDTA is one of the most widely used chelating agents and is very effective in promoting phytoremediation of Cd, Pb, and other heavy metal-contaminated soils by enhancing the mobility of heavy metals in soil, it has been suggested that EDTA is a potential risk to groundwater and drinking water due to its remobilization of metals from sediments and soil [39]. To reduce the potential environmental risk of EDTA, a series of easily biodegradable chelating agents, including citric acid (CA), sodium glutamate tetraacetate (GLDA), oxalic acid (OA), and diethylenetriaminepentaacetic acid (DTPA), were used in this study.

The chelating agent GLDA, made from renewable plant materials, is readily biodegradable and comparable to EDTA in terms of chelating capacity. By binding to heavy metal ions and forming chelates, GLDA can activate heavy metal ions in the soil, increase their bioavailability, reduce their toxicity to plants, and promote the uptake and aerial transport of heavy metals by plants [40]. A mixture of GLDA–ascorbic acid is also a potential candidate for Pb and Zn removal, as it removes approximately 90% of Pb and 70% of Zn [41]. Previous studies have shown that within one month, more than 60% of GLDA undergoes degradation [42,43]. Therefore, it has the lowest ecological footprint when compared to other chelating agents, such as EDTA and sodium tripolyphosphate (STPP) [44,45]. Hence, this biodegradable chelator may provide a good and reasonably cost-effective solution to treat contaminated soils.

Most phytoremediation experiments are carried out on plants with hyperaccumulation of heavy metals. However, most of these species are not suitable for commercial phytoremediation, mainly due to their low annual harvestable biomass and low growth rate [46]. Ryegrass, a genus of the Poaceae, is a heavy metal-enriched plant and a common cool-season turfgrass for lawn establishment in northern China owing to its rapid growth, well-developed root system, large biomass, and strong adaptability. It has good potential for certain heavy metals and is suitable for repairing heavy metals in soil. Ryegrass also shows good potential in phytoextraction for single metal Cd pollution and combined Cd and Zn pollution [47].

In this paper, pot experiments were conducted using ryegrass and artificially contaminated soil. The effect of chelating agents (EDTA, GLDA, DTPA, OA, and CA) on the phytoremediation of heavy metals Cd and Pb by ryegrass was studied by measuring soil physiochemical properties, ryegrass biomass, and the contents of heavy metals in soil and ryegrass. The extraction potential of chelating agents for heavy metals in soil was explored using correlation analysis. This research aimed to provide precise technical parameters for Cd- and Pb-polluted soil phytoremediation.

## 2. Materials and Methods

### 2.1. Soil and Plant Sample Analysis

The concentrations of heavy metal fractions in the soil under study were determined using the sequential extraction method of the European Community Reference Bureau (BCR) [48]. The four fractions were categorized as follows: Step 1—acid extractable fraction; Step 2—reducible fraction (bound to Fe-Mn oxides); Step 3—oxidizable fraction (bound to organic matter and sulfides); Step 4—residue fraction. To determine the total concentration of Cd and Pb in the soil samples, 0.5 g of the sample was digested with a mixture of acids (9 mL of concentrated $HNO_3$ + 3 mL of concentrated HCl) in a microwave unit (MDS-6G, Sineo, Shanghai, China), according to EPA method 3051A. The concentration of each fraction was analyzed using an atomic absorption spectrometer (Zeenit700P, Analytik Jena, Jena, Germany).

For plant sample analysis, 0.1 g of the plant sample was placed in a 50 mL Polytetrafluoroethylene (PTFE) crucible, and 5 mL of HCl was added for initial digestion, followed by addition of $HNO_3$, $HClO_4$, and HF. The crucible was covered and heated on an electric hot plate. After digestion, the solution was transferred to a 25 mL volumetric flask, and 3 mL of hydrogen phosphate diamine solution was added. The contents of Cd and Pb were determined using a Flame Atomic Absorption Spectrometer (Zeenit700P, Analytik Jena, Jena, Germany). The effects of different types and concentrations of chelating agents on plant growth, biomass, uptake of heavy metals by different parts of the plant, and the impact of chelator addition on the extractable state of heavy metals in the rhizosphere and non-rhizosphere soils were analyzed and compared. Three parallel samples were taken for each treatment.

### 2.2. Sample Collection and Experimental Processing

2.2.1. Sample Collection

The soil samples used in this study were collected from the surface layer (0–20 cm depth) of agricultural land located in the southern suburbs of Xi'an, Shaanxi Province, China, at coordinates 34°9′24″ N, 108°46′1″ E. Over 60% of the study area is classified as Class II–III self-weighting wet sinking loess, and the soil is primarily composed of brown and Lou soil, which are characterized by a thicker texture and better permeability. Ryegrass seeds were purchased from Hundred Green Landscape Design Company in Xi'an, Shaanxi Province, China. Table 1 presents the basic physicochemical properties of the study area.

**Table 1.** Basic physicochemical properties of soils in the study area.

| Index | $pH_{1/2.5}$ | Organic Material (g·kg$^{-1}$) | Total Potassium (mg·kg$^{-1}$) | Total Cd (mg·kg$^{-1}$) | Total Pb (mg·kg$^{-1}$) |
|---|---|---|---|---|---|
| Values | 7.70 | 13.80 | 0.31 | 0.33 | 16.3 |

2.2.2. Analysis of the Effect of Different Chelating Agents on Heavy Metal Extraction in the Study Soil

A total of 4.0 g of dry soil sample was placed into a 50 mL centrifuge tube and five different types of chelating agents, including CA, DTPA, EDTA, GLDA, and OA, were added at concentrations of 5, 10, and 20 mmol·L$^{-1}$ (when the concentration of chelating

agents is 5, 10, and 20 mmol·kg$^{-1}$, the CA treatment groups were named C5, C10, and C20. Similarly, the treatment groups of EDTA, GLDA, DTPA, and OA were named E5, E10, E20, G5, G10, G20, D5, D10, and D20 alongside O5, O10, and O20. Ultrapure water was used as the control. In total, 16 treatments were performed with 5 parallel samples for each treatment. After treatment, 40 mL of the chelating agents were added to the soil samples and shaken at 200 rpm for 10 h, followed by centrifugation at 3000 rpm for 20 min. The supernatants were collected, filtered through a 0.45 μm filter, and stored at 4 °C. The concentrations of Cd and Pb in the chelating agent extraction solution were measured, and the contents of different chemical combinations of Cd and Pb in the soil were determined using the BCR continuous extraction method.

### 2.2.3. Analysis of the Effect of Different Chelating Agents on Heavy Metal Extraction in Ryegrass

In the laboratory, soil samples were artificially contaminated with lead nitrate Pb(NO$_3$)$_2$ (400 mg·kg$^{-1}$) and cadmium nitrate Cd(NO$_3$)$_2$ (10 mg·kg$^{-1}$). Then, the samples were passivated for two weeks. After air-drying, the samples were potted and planted with ryegrass. Each pot contained 50 sterilized ryegrass seeds and 1 kg of artificially polluted soil, which was maintained at 60% water-holding capacity in the field. After one week, when the seedlings reached a height of 3–4 cm, they were interrupted, and only 30 seedlings were left in each pot. The pots were placed in a simulated greenhouse at a constant temperature of approximately 25 °C. The positions of the pots were changed every 10 days to ensure even light distribution. After 45 days of growth, chelating agents, including CA, EDTA, and GLDA, were dissolved in water at concentrations of 2, 4, and 6 mmol·kg$^{-1}$. When the concentration of chelating agents was 2, 4, or 6 mmol·kg$^{-1}$, the CA treatment groups were named C2, C4, and C6, respectively. Similarly, the treatment groups of EDTA and GLDA were named E2, E4, E6, and G2, G4, G6. Then, the chelating agents were showered on the soil surface. The growth status of the plants was observed and recorded. After 15 days, the plants were harvested, and the aerial part was cut at a height of 1 cm from the soil surface to be used as the aboveground part. Then, the soil block was gently broken to harvest the roots as the belowground part. The different parts of ryegrass were rinsed repeatedly with tap water, three times with ultrapure water, and wiped with absorbent paper. The height of the aerial part and the length of the root system were measured. The aerial and root parts were, then, placed in separate bags, incubated at 105 °C in an oven for 30 min to kill any microorganisms, and dried at 80 °C to a constant weight. The dry weights of the aerial and root parts were determined. Finally, the samples were crushed, passed through a 100-mesh sieve, and stored in sealed bags.

### 2.3. Data Processing

The data obtained from the measured samples were analyzed using SPSS 20.0 software, and Pearson and Spearman correlation analyses were conducted on the data. The results of the pot experiments are presented as the mean ± standard deviation of three replicates. Significant differences ($p < 0.05$) were determined using Duncan's method and the data were analyzed and plotted using Origin 8.1.

The correlation coefficient was calculated as follows:

$$BCF = \frac{Cp}{Cs} \tag{1}$$

$$TF = \frac{Ca}{Cr} \tag{2}$$

$$HMEE = \frac{Tp}{Ts} \tag{3}$$

where *BCF*: bioaccumulation factor; *Cp*: heavy metal concentrations in aerial and root parts of plants, mg·kg$^{-1}$ dry weight; *Cs*: heavy metal concentrations in soil, mg·kg$^{-1}$;

*TF*: translocation factor; *Ca*: heavy metal concentrations in aerial parts of plants, mg·kg$^{-1}$; *Cr*: heavy metal concentrations in root parts of plants, mg·kg$^{-1}$; *HMEE*: heavy metals extraction efficiency, %; *Tp*: total plant-derived heavy metals, mg; *Ts*: total heavy metals in soil, mg.

## 3. Results and Discussion

### 3.1. Comparison of the Activation Effect of Chelating Agents on Cd and Pb in Soil

The extraction efficiencies of different chelating agents, including EDTA, GLDA, DTPA, OA, and CA on Cd in the study soil are shown in Figure 1A, and Pb in Figure 1B.

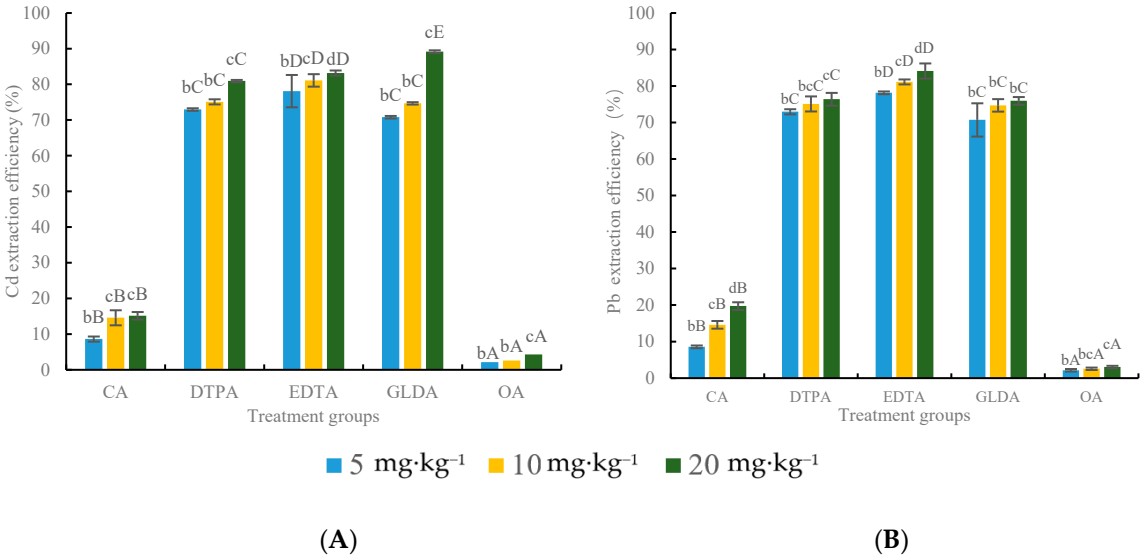

**(A)**                                                    **(B)**

**Figure 1.** Extraction efficiencies of different types of chelating agents for heavy metals Cd and Pb in soil. Note: The analysis results were obtained using a two-factor ANOVA. Lowercase letters denote significant differences ($p < 0.05$) in the extracted amounts of Cd or Pb between different concentration levels of the same chelating agent. Different uppercase letters denote significant differences ($p < 0.05$) in the extracted amounts of Cd or Pb at the same concentration level between different chelating agent species. The standard deviation is based on 18 samples.

Different concentrations of five chelators, including CA, DTPA, EDTA, GLDA, and CA, in a range from 5 mg·kg$^{-1}$ to 20 mg·kg$^{-1}$, were tested in order to study the effect of single chelating agent concentrations on the extraction of heavy metals (Figure 1). Unsurprisingly, it was found that higher concentrations (20 mg·kg$^{-1}$) of chelating agents, especially of GLDA led to an increase in heavy metal removal efficiency.

Figure 1A shows that different chelating agents exhibited varying extraction efficiencies for Cd in the soil. The tested chelating agents were ranked in descending order of extraction efficiency as follows: GLDA > EDTA > DTPA > CA > OA. The APCA-class chelating agents, namely EDTA, GLDA, and DTPA, exhibited the highest extraction efficiencies for Cd (greater than 67.67%). In contrast, the low molecular weight organic acid chelator CA had a medium extraction efficiency, with the largest extraction efficiency of 15.25%. The extraction efficiency of OA for Cd was the lowest at less than 5.0%. Overall, APCAs demonstrated significantly higher extraction efficiencies for Cd than low molecular weight organic acid chelating agents.

Regarding the extraction of Pb in the study soil, Figure 1B demonstrates that different chelating agents had varying extraction efficiencies. The chelating agents were ranked in descending order of extraction efficiency as follows: EDTA > DTPA > GLDA > CA > OA. The extraction efficiencies of APCA-class chelating agents for Pb ranged from 74.17% to 84.1%. The highest extraction efficiency of the low molecular organic acid chelator CA for Pb was 19.80%. In contrast, the extraction efficiency of OA for Pb was only 2.8%, which

was close to the value of the control group (1.18%). Overall, the tested APCAs (EDTA, DTPA, and GLDA) demonstrated significantly higher extraction efficiencies for Pb than CA and OA. Furthermore, the extraction efficiencies increased with increasing chelating agent concentrations until 10 mmol·kg$^{-1}$. However, concentrations greater than 10 mmol·kg$^{-1}$ inhibited plant growth and the absorption of heavy metals.

The solubility of complex metal-extractant is the key issue in the washing of contaminated soils, which directly influences the metal removal efficiency of chelating agents. Depending on their dissolution constants and a number of carboxylic groups, organic chelating agents may carry a varying negative charge, which has a high affinity for metal ions. In solution, the equilibrium speciation in a metal-extractant complex is controlled by the media pH, concentration of all metals, and chelating agents, as well as the stability constants of the complexes (log K value) [42].

### 3.2. Effect of Chelating Agents on the Speciation of Cd and Pb in Soil

The distribution pattern of heavy metal species in the soil is closely related to their mobility and bioavailability in the soil [49,50]. Therefore, understanding the distribution of heavy metal forms in the soil is essential in selecting suitable chelating agents and plants for remediation purposes.

Figure 2 illustrates the effect of chelator leaching on the morphology of residual Cd in soil. The results showed that before the addition of chelating agents, the dominant form of Cd in the soil was the weak acid extractable state, which accounted for 72.31% of the total Cd. EDTA, GLDA, and DTPA had the most significant effect on the extraction of heavy metals from the weak acid extractable state in the soil, with the Cd content in the weak acid state reduced by 75.81–85.64%, 80.85–89.13%, and 87.45–95.99%, respectively, compared to the control group (CK). Although CA facilitated the extraction of Cd in the weak acid extractable state in the soil, its effect was not significant compared to the APCA-class of chelating agents. In the OA treatment group, no significant change was observed in the Cd content of various forms in the soil under study. The application of chelators reduced the content of Cd in the reducible and oxidizable states, although it had little effect on the residual state of Cd.

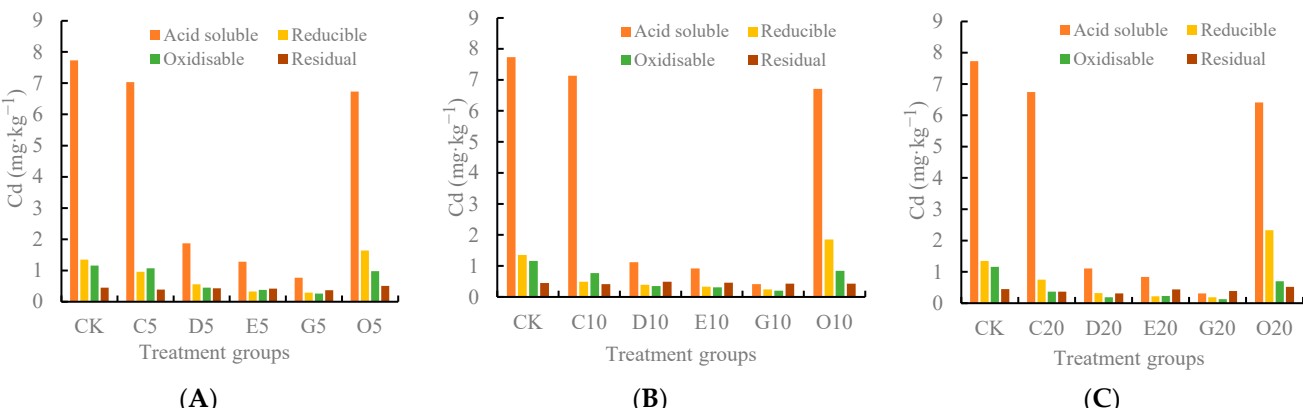

**Figure 2.** Changes in speciation of heavy metal Cd in the soil after leaching with chelating agents. Note: (**A**) Changes in speciation of Cd in the soil after leaching when the concentration of chelating agent is 5 mmol·kg$^{-1}$; (**B**) Changes in speciation of Cd in the soil after leaching when the concentration of chelating agent is 10 mmol·kg$^{-1}$; (**C**) Changes in speciation of Cd in the soil after leaching when the concentration of chelating agent is 20 mmol·kg$^{-1}$. ① CK: purified water 0 mmol·kg$^{-1}$; ② the standard deviation is from 18 samples, the same as below.

Figure 2 is the Changes in speciation of heavy metal Cd in the soil after leaching with different concentration of chelating agents, 5 mmol·kg$^{-1}$ (Figure 2A), 10 mmol·kg$^{-1}$ (Figure 2B), and 20 mmol·kg$^{-1}$ (Figure 2C). It can be seen that under the three concentrations of chelating agents, the concentration changes of various forms of Cd are not

significant, indicating that the concentration of chelating agents has no significant impact on the extraction efficiency of different forms of Cd in soil.

Figure 2 depicts the dominant chemical forms of Pb in the studied soil, which were weak acid extractable state and reducible state, accounting for 95.19% of the total Pb in the soil. The three chelating agents, namely EDTA, GLDA, and DTPA, had a significant effect on the extraction of Pb from the weakly acidic and reducible states. The Pb content of the weakly acidic state was reduced by 84.62–86.00%, 73.80–77.02%, and 75.82–79.41%, respectively, compared to the control group (CK). Similarly, Pb in the reducible state was reduced by 77.31–86.03%, 69.71–80.49%, and 72.33–76.80%, respectively, compared to the CK. However, the application of these chelating agents had little effect on the residual state of Pb in the soil, which is consistent with previous research [51,52]. The high mobility of the weakly acidic extractable and reducible forms of Pb may explain this finding since these forms are easily released under acidic and reducing conditions, whereas the oxidizable and residual forms are primarily found in humus and mineral particles or within the lattice of silicates, primary, and secondary minerals in the soil, which are not easily extracted. CA showed effectiveness in extracting both forms of Pb from the soil, although the effect was not significant compared to the APCA-class of chelating agents, and it was also less efficient at extracting residual Pb. The reason for the low extraction efficiency of residual metals in soil by chelating agents is that residual Cd and Pb were tightly bound to the lattice of minerals (e.g., silicates) in the soil and, therefore, difficult to extract even by chemical washing with chelating agents [53].

Figure 3 is the Changes in speciation of heavy metal Cd in the soil after leaching with different concentration of chelating agents, 5 mmol·kg$^{-1}$ (Figure 3A), 10 mmol·kg$^{-1}$ (Figure 3B), and 20 mmol·kg$^{-1}$ (Figure 3C). It can be seen that under the three concentrations of chelating agents, the concentration changes of various forms of Pb are not significant, indicating that the concentration of chelating agents has no significant impact on the extraction efficiency of different forms of Pb in soil.

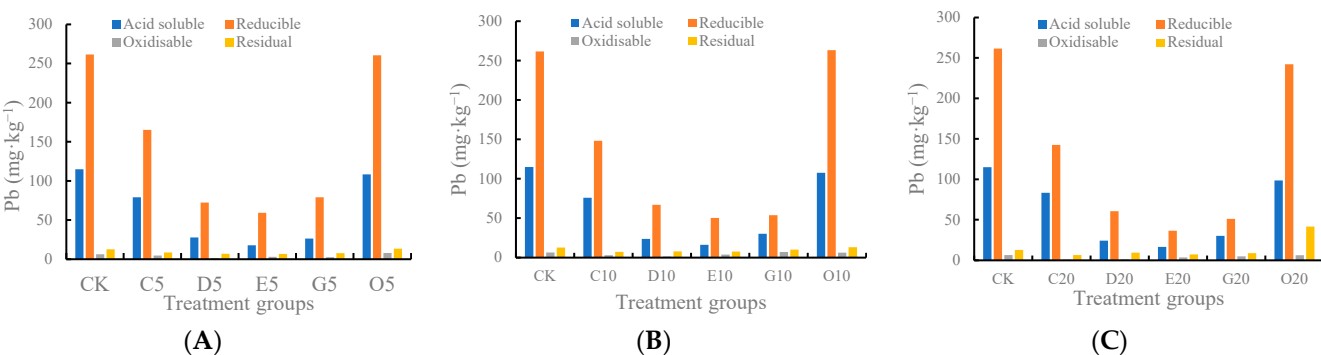

**Figure 3.** Changes in speciation of Pb in the soil after leaching with chelating agents. Note: (**A**) Changes in speciation of Pb in the soil after leaching when the concentration of chelating agent is 5 mmol·kg$^{-1}$; (**B**) Changes in speciation of Pb in the soil after leaching when the concentration of chelating agent is 10 mmol·kg$^{-1}$; (**C**) Changes in speciation of Pb in the soil after leaching when the concentration of chelating agent is 20 mmol·kg$^{-1}$.

The number of ligand atoms in a chelating agent determines its ability to complex metals in the soil [54]. The five chelating agents used in this study, namely DTPA, EDTA, GLDA, CA, and OA, provided 8, 6, 5, 4, and 2 coordination atoms, respectively. The chelating agent OA also formed insoluble compounds such as Cd oxalate and Pb oxalate. In addition to having a lower number of coordination atoms, this characteristic resulted in a weaker activation effect of OA on the studied metals (Cd and Pb). EDTA, GLDA, and DTPA had the most significant impact on the extraction of Pb from the weakly acidic and reducible states of the soil, with the content of Pb in the weakly acidic state reduced by 84.62–86.00%, 73.80–77.02%, and 75.82–79.41%, respectively.

After the addition of the chelating agents, heavy metal ions in the solid phase of the soil can be transformed into a soluble state, thereby enhancing their absorption by plants [55]. As shown in Figures 2 and 3, the same chelating agent exhibits varying extraction efficiencies for different heavy metals in the soil. For instance, the extraction efficiency of EDTA for Pb is higher than that for Cd, which is mainly determined by the stability constant logK of the chelating agent [56,57].

The extraction efficiency of the chelating agents for heavy metals in soil depends on several factors, such as the chelator's structure, the solubility of the extractant, the type of heavy metals, and the soil's physicochemical properties, etc., while the solubility of the extractant is the key issue in the washing of contaminated soils, which directly influences the metal removal efficiency of the chelating agents. Depending on their dissolution constants and a number of carboxylic groups, organic chelating agents may carry a varying negative charge, which has a high affinity for metal ions. In solution, the equilibrium speciation in a metal-extractant complex is controlled by the media pH, concentration of all metals, and chelating agents, as well as the stability constants of the complexes (log K value). Cd is known to exist in the forms of carbonates, hydroxides, and phosphates, and their solubility and availability increase at a low soil pH, while soil acidity can increase the desorption of Cd in soil colloids, facilitating its uptake by roots.

APCA-class chelating agents were effective in reducing heavy metal content in the weak acid extractable and reducible states in the soil, although they were less effective in extracting heavy metals in the residue state. Chelating agents increased the proportion of heavy metals in the residual state in the soil and reduced their harmful effects on the soil ecosystem. Biodegradable chelating agents such as GLDA, DTPA, and CA have great potential in enhancing the phytoremediation of Cd- and Pb-contaminated soils.

Our experimental results demonstrated a phenomenon in which the removal efficiency for Cd and Pb is low. Moreover, the study results of Fe-lili, et al. showed that after growing ryegrass, the contents of heavy metals decreased by 14% for Cd and 44% for Pb; after spraying EDTA it decreased again by 24% for Cd, and 68% for Pb. In ryegrass, the uptake of heavy metals was enhanced, and the bioconcentration factor of Cd, and Pb in EDTA-treated groups was 2.7 and 4.8 times the control group, respectively. Thus, in the following pot experiments, CA and OA were not added.

*3.3. Effect of Chelating Agents on the Amount of Extracted Heavy Metals in Ryegrass*

Although CA has been favored by researchers due to its high biodegradability and its ability to stimulate metal uptake in plants [58]. CA also has adverse effects on plant growth and development, as it can cause etiolation, withering, and even death when present in excess amounts [59].

Plant growth plays a critical role in the phytoextraction of heavy metals from the contaminated soil. The addition of chelating agents could increase the root biomass. Figure 4A illustrates that EDTA, GLDA, and CA significantly increased the uptake of Cd in both the aerial and root parts of ryegrass compared to the control group (CK). As the concentration of EDTA and GLDA increased, the amount of Cd extracted from the aerial fraction continued to decrease. At 2 mmol·kg$^{-1}$, the extraction amount of Cd in the aerial part was relatively high, reaching 19.18 μg Cd·pot$^{-1}$, and 24.79 μg Cd·pot$^{-1}$, respectively, which were 2.33 and 3.30 times higher than the CK. However, the results obtained from CA and the EDTA/GLDA mixture were different. As the concentration of CA increased, the amount of Cd extracted from the aerial parts increased, reaching a maximum of 10.11 μg Cd·pot$^{-1}$ when the concentration of CA was 6 mmol·kg$^{-1}$, which was 0.76 times higher than the CK. This is because high concentrations of EDTA and GLDA have a toxic effect on plants and inhibit their growth and development, while CA is less toxic and can promote plant growth.

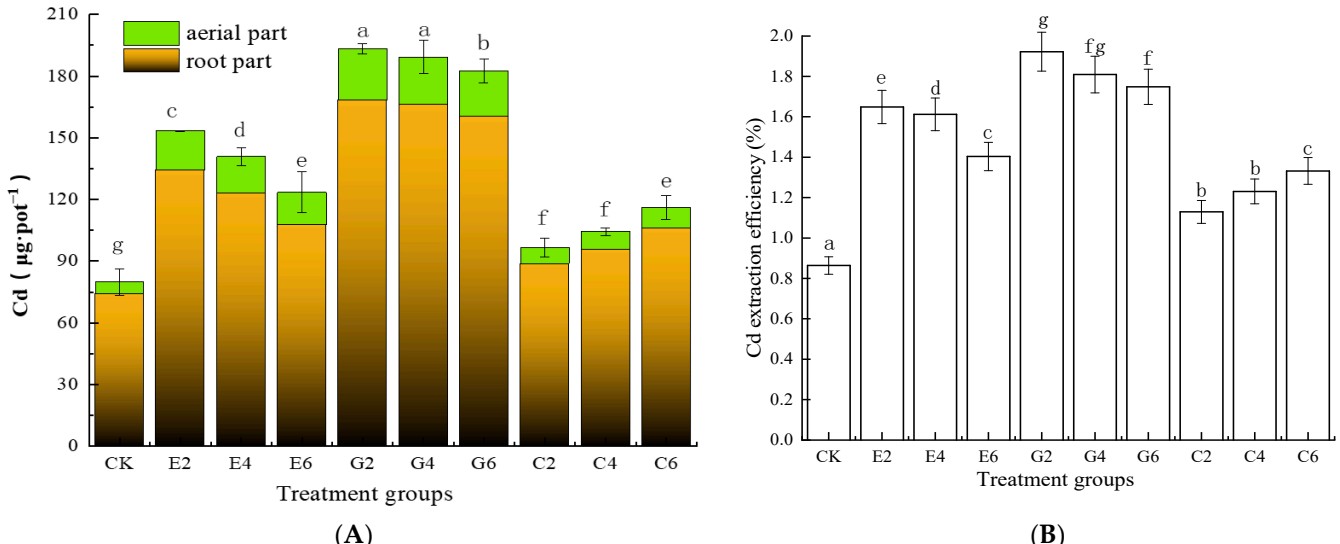

**Figure 4.** Effect of chelating agents on total Cd extraction quantity and efficiency in ryegrass. Note: This is the analysis result using one-way ANOVA, and the different lowercase letters denote the significant differences between two groups ($p < 0.05$). The standard deviation is from 10 samples.

Regarding the extraction from the root fraction, the extraction decreased with increasing concentrations of EDTA and GLDA. At 2 mmol·kg$^{-1}$, the maximum extraction of Cd was 134.21 μg Cd·pot$^{-1}$ in the root part and 168.52 μg Pb·pot$^{-1}$ in the aerial part, respectively, which were 0.81 and 1.27 times higher than the CK. Finally, the root part extraction under CA treatment increased continuously with increasing CA concentrations. A maximum of 106.08 μg·pot$^{-1}$ was reached with 6 mmol·kg$^{-1}$ of CA, which was 0.43 times higher than the CK. The extraction order was GLDA > EDTA > CA at the same chelator concentrations. Moreover, the extraction enrichment of soil Cd by ryegrass was mainly in the roots, accounting for more than 85% of the total plant Cd.

In Figure 4B, among the three different types of treatment groups, GLDA had the highest extraction efficiency ranging from 1.83% to 1.93% when removing Cd from ryegrass, which was 1.28 to 1.42 times higher than the CK. The EDTA-treated group had the second-highest extraction efficiency, with a range of 1.24% to 1.53%, which was 0.54 to 0.92 times higher than the CK. CA was the least effective in promoting the extraction of Cd from the soil, with an extraction efficiency ranging from 0.97% to 1.16%, which was only 0.21 to 0.45 times higher than the CK.

The results presented in Figure 5A indicated that EDTA, GLDA, and CA significantly increased the uptake of Pb in both the aerial and root parts of ryegrass compared to the CK. As the concentrations of EDTA and GLDA increased, the aerial extraction of Pb from ryegrass gradually decreased. The maximum extraction of Pb from the soil by the aerial portion with 2 mmol·kg$^{-1}$ of EDTA was 0.62 mmol·kg$^{-1}$, which was 2.88 times higher than the CK. The maximum extraction of Pb by the aerial parts of ryegrass was 650 mg Pb·pot$^{-1}$ with 2 mmol·kg$^{-1}$ of GLDA, which was 3.06 times higher than the CK. The extraction of Pb increased continuously with increasing CA concentrations. The maximum extraction of Pb was 0.35 mg Pb·pot$^{-1}$ with 6 mmol·kg$^{-1}$ of CA, which was 1.18 times higher than the CK.

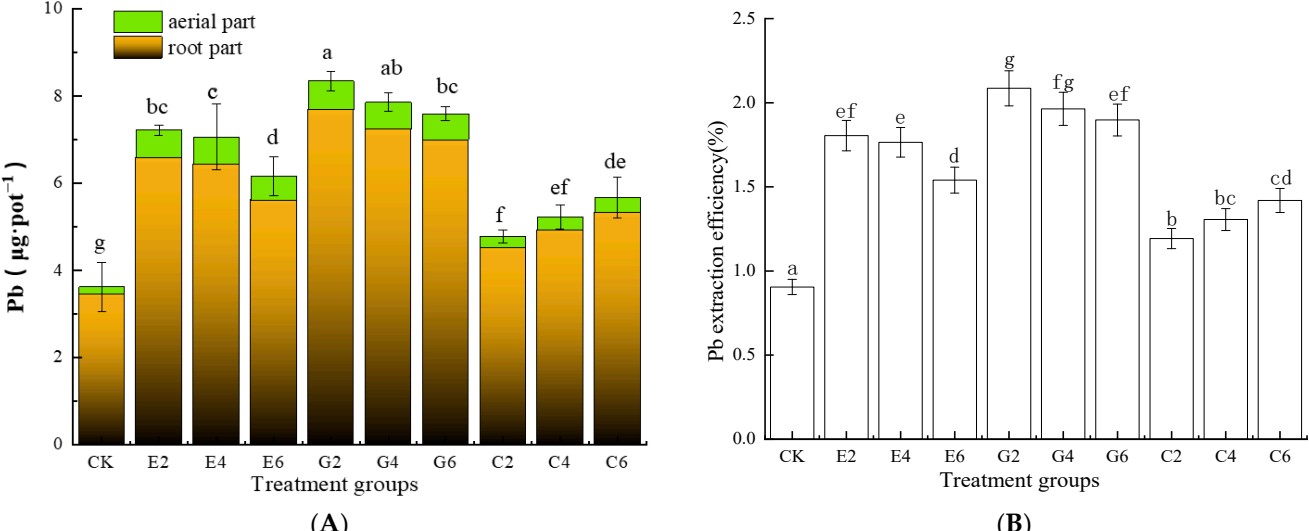

**Figure 5.** Effect of chelating agents on total Pb extraction quantity and efficiency in ryegrass ($p < 0.05$). Note: This is the analysis result using one-way ANOVA, and the different lowercase letters denote the significant differences between two groups ($p < 0.05$). The standard deviation is from 10 samples.

In contrast, the extraction of Pb in the root part decreased with increasing concentrations of the chelating agents. When the concentration of EDTA and GLDA was 2 mmol·kg$^{-1}$, the maximum extraction of Pb was 6.59 mg Pb·pot$^{-1}$ in the root part and 7.70 mg Pb·pot$^{-1}$ in the aerial part, which was 0.9 and 1.22 times higher than the CK, respectively. Conversely, the root extraction of CA increased with increasing concentration of CA. The maximum extraction was 5.33 mg Pb·pot$^{-1}$ with 6 mmol·kg$^{-1}$ of CA, which was 0.54 times higher than the CK. When using the same concentrations of chelating agents, the extraction effect was in the following descending order: EDTA > GLDA > CA. The ryegrass roots enriched more than 91% of the plant Pb.

In Figure 5B, GLDA exhibited the highest extraction efficiency for Pb from ryegrass among the three treatment groups, ranging from 1.90 to 2.09%, which was 1.11 to 1.32 times higher than that of CK. The EDTA-treated group followed, with an extraction efficiency of Pb from ryegrass ranging from 1.54 to 1.80%, which was 0.71 to 1.00 times higher than the CK. However, CA was the least effective at extracting Pb from the soil, with an extraction efficiency ranging from 1.19 to 1.42%, which was only 0.33 to 0.58 times higher than the CK.

The uptake of soil Cd/Pb by ryegrass was determined by the biomass of ryegrass and the concentration of heavy metals in the plant. Although the application of chelating agents reduced the aerial and root biomass of ryegrass, the concentrations of Cd and Pb in the plant increased, thereby increasing the total uptake of Cd/Pb by the ryegrass. It was also observed that although the addition of chelating agents promoted heavy metal enrichment and transport in ryegrass, the enrichment of Cd and Pb in ryegrass was still mainly in the root part. This is consistent with a previous study where Cd and Pb were mainly concentrated in the roots rather than in the aerial part, indicating that the root system is the primary organ for Cd and Pb storage in ryegrass [60].

### 3.4. Correlation Analysis

The introduction of chelating agents impacted both the biomass of the ryegrass and the accumulation of heavy metals in different parts of the plant. The results of the correlation analyses are presented in Tables 2–4.

**Table 2.** Correlation analysis of EDTA in various parts of ryegrass.

| Items | EDTA | Aerial Dry Weight | Root Dry Weight | Aerial (Pb) | Root (Pb) | Aerial (Cd) | Root (Cd) | Total Extraction of Pb | Total Extraction of Cd |
|---|---|---|---|---|---|---|---|---|---|
| EDTA | 1.00 | | | | | | | | |
| Aerial part biomass (dry weight) | −0.823 ** | 1.00 | | | | | | | |
| Root part biomass (dry weight) | −0.946 ** | 0.834 ** | 1.00 | | | | | | |
| Aerial part (Pb) | 0.737 * | −0.865 ** | −0.707 * | 1.00 | | | | | |
| Root part (Pb) | 0.884 ** | −0.916 ** | −0.814 ** | 0.917 ** | 1.00 | | | | |
| Aerial part (Cd) | 0.691 * | −0.683 * | −0.23 | 0.65 | 0.63 | 1.00 | | | |
| Root part (Cd) | 0.793 * | −0.955 ** | −0.758 * | 0.950 ** | 0.956 ** | 0.730 * | 1.00 | | |
| Total extraction of Pb | −0.689 * | 0.49 | 0.828 ** | −0.26 | −0.36 | 0.21 | −0.33 | 1.00 | |
| Total extraction of Cd | −0.919 ** | 0.66 | 0.931 ** | −0.46 | −0.668 * | −0.04 | −0.55 | 0.862 ** | 1.00 |

Note: * $p < 0.05$, ** $p < 0.01$ (two-tailed).

**Table 3.** Correlation analysis of GLDA in various parts of ryegrass.

| Items | GLDA | Aerial Dry Weight | Root Dry Weight | Aerial (Pb) | Root (Pb) | Aerial (Cd) | Root (Cd) | Total Extraction of Pb | Total Extraction of Cd |
|---|---|---|---|---|---|---|---|---|---|
| GLDA | 1 | | | | | | | | |
| Aerial part biomass (dry weight) | −0.874 ** | 1 | | | | | | | |
| Root part biomass (dry weight) | −0.931 ** | 0.947 ** | | | | | | | |
| Aerial part (Pb) | 0.719 * | −0.878 ** | −0.839 ** | 1 | | | | | |
| Root part (Pb) | 0.934 ** | −0.980 ** | −0.969 ** | 0.873 ** | | | | | |
| Aerial part (Cd) | 0.710 * | −0.768 * | −0.58 | 0.792 * | 0.697* | 1 | | | |
| Root part (Cd) | 0.904 ** | −0.954 ** | −0.945 ** | 0.908 ** | 0.974** | 0.698 * | 1 | | |
| Total extraction of Pb | −0.873 ** | 0.794 * | 0.935 ** | −0.67 | −0.828 ** | −0.30 | −0.804 ** | 1 | |
| Total extraction of Cd | −0.66 | 0.38 | 0.61 | −0.18 | −0.49 | 0.16 | −0.38 | 0.759 * | 1 |

Note: * $p < 0.05$, ** $p < 0.01$ (two-tailed).

**Table 4.** Correlation analysis of CA in various parts of ryegrass.

| Items | CA | Aerial Dry Weight | Root Dry Weight | Aerial (Pb) | Root (Pb) | Aerial (Cd) | Root (Cd) | Total Extraction of Pb | Total Extraction of Cd |
|---|---|---|---|---|---|---|---|---|---|
| CA | 1.00 | | | | | | | | |
| Aerial dry weight | −0.54 | 1.00 | | | | | | | |
| Root dry weight | −0.771 * | 0.788 * | 1.00 | | | | | | |
| Aerial (Pb) | 0.946 ** | −0.66 | −0.782 * | 1.00 | | | | | |
| Root (Pb) | 0.842 ** | −0.821 ** | −0.850 ** | 0.947 ** | 1.00 | | | | |
| Aerial (Cd) | 0.697 * | −0.835 ** | −0.857 ** | 0.827 ** | 0.957 ** | 1.00 | | | |
| Root (Cd) | 0.933 ** | −0.772 * | −0.849 ** | 0.949 ** | 0.948 ** | 0.868 ** | 1.00 | | |
| Total extraction of Pb | 0.812 ** | −0.751 * | −0.714 * | 0.942 ** | 0.975 ** | 0.910 ** | 0.910 ** | 1.00 | |
| Total extraction of Cd | 0.906 ** | −0.722 * | −0.734 * | 0.940 ** | 0.931 ** | 0.843 ** | 0.979 ** | 0.937 ** | 1.00 |

Note: * $p < 0.05$, ** $p < 0.01$ (two-tailed).

Tables 2 and 3 reveal a significant negative correlation between the concentrations of EDTA and GLDA and the dry weight of both the aerial and root parts of ryegrass. Conversely, there was a significant positive correlation between the concentrations of EDTA and GLDA and the concentrations of Cd and Pb in ryegrass, indicating that higher concentrations of chelating agents led to higher concentrations of Cd and Pb in ryegrass. However, there was a significant negative correlation between the concentrations of EDTA and GLDA and the total enrichment of Cd and Pb in ryegrass, suggesting that higher concentrations of chelating agents led to less effective enrichment of Cd and Pb in ryegrass from the soil.

It can be observed that there was a significant negative correlation between the concentration of CA and the dry weight of the aerial and root parts of ryegrass, indicating that higher CA concentrations resulted in a decrease in the dry weight of these plant

parts. Moreover, the concentration and total extraction of Cd and Pb in ryegrass increased continuously with increasing concentrations of CA, implying that the higher the applied concentration of CA, the more effective the enrichment of Cd and Pb in the soil by ryegrass.

## 4. Conclusions

In this study, five chelating agents, including GLDA, EDTA, DTPA, OA, and CA, were compared for their efficacy in enhancing the phytoremediation of Cd and Pb in artificially contaminated soil. The following conclusions were drawn:

(1) The leaching experiments using biodegradable chelating agents indicated that the order of Cd removal effectiveness was: GLDA > EDTA > DTPA > CA > OA removal effectiveness, while the order of Pb was: EDTA >DTPA > GLDA > CA > OA.

(2) The total extraction of Cd from the study soil by chelating agents of CA, EDTA, and GLDA ranged from 116.19 to 193.31 µg Cd·pot$^{-1}$ and 5.88 to 8.35 mg Pb·pot$^{-1}$ for Pb. CA, EDTA, and GLDA had a range of 0.97–1.93% for Cd and 1.19–2.09% for Pb extraction efficiency under the same conditions. When the GLDA concentration was 2 mmol·kg$^{-1}$, the extraction of Cd and Pb was the best, which was 1.42 and 1.32 times higher than the untreated group. When chelator concentrations were higher than 10 mmol·kg$^{-1}$, the growth of ryegrass and the absorption of Cd and Pb were inhibited.

(3) When the concentration of EDTA, GLDA, and CA was 6 mol·L$^{-1}$, the Pb concentration in the aerial part of ryegrass ranged from 67.08 to 139.44 mg·kg$^{-1}$ and 442.33 to 834.91 mg·kg$^{-1}$ in the root part. The Cd concentration in the aerial part ranged from 1.95 to 4.49 µg·kg$^{-1}$ and 8.8 to 17.79 µg·kg$^{-1}$ in the root part. The heavy metals concentrations in ryegrass increased with the application of the chelating agents, with the order of enrichment being GLDA > EDTA > CA for Cd, and EDTA > GLDA > CA for Pb.

This study utilized indoor pot experiments with several artificial control factors, such as light, temperature, and soil water holding capacity. However, it is challenging to ensure that these factors remained constant during the actual restoration process. Therefore, future research needs to consider field-scale phytoremediation of soil. In addition to the performed experiments, simultaneous phytoremediation experiments in situ on heavy metal-contaminated soils should also be considered.

**Author Contributions:** Conceptualization, resources, W.D.; writing—original draft preparation, W.D. and R.W.; writing—review and editing, W.D. and H.W.; data curation, R.W.; supervision, H.L. and J.L.; formal analysis, X.Y.; investigation, W.D., R.W., C.J., H.W. and Z.W. All authors have read and agreed to the published version of the manuscript.

**Funding:** This research was funded by the National Natural Science Foundation of China (No.51879215; 52000150); the Natural Science Foundation of Shaanxi Province, China (2021JM-329).

**Data Availability Statement:** Data is contained within this article. The data that support the findings of this study are available from the corresponding author, [W.D], upon reasonable request.

**Conflicts of Interest:** The authors declare no conflict of interest.

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
