# Peer review of "Effects of Chelating Agents Addition on Ryegrass Extraction of Cadmium and Lead in Artificially Contaminated Soil"

_water, doi:10.3390/w15101929_

Round 1

Reviewer 1 Report

This work mainly studied the effects of the growth of ryegrass combined with chelating agents on the removals of Cd and Pb in the contaminated soils. The authors found that the addition of some chelating agents can reduce the content of available Cd and Pb forms in the soils, but is less effective to change their residual forms. The applications of chelating agents promote the uptakes of Cd and Pb by ryegrass, but reduce its biomass. In general, the results obtained from the pot experiments are reliable. But the English is poorly written with many grammar mistakes, and should be completely revised.

The English expression of the whole manucript should be completely rewritten. I only point out some problems as follows:

Line 28: “the in so”?

Line 35-36: grammar mistake.

Line 37-38: bad expression.

Line 411-444: the conclusions should be shortened.

Author Response

Thank you so much for your valuable comments. We are sorry to keep you waiting. We have carefully revised the article and improved the discussion section based on your good suggestions and requests, and marked it in red in the manuscript. 

This work mainly studied the effects of the growth of ryegrass combined with chelating agents on the removals of Cd and Pb in the contaminated soils. The authors found that the addition of some chelating agents can reduce the content of available Cd and Pb forms in the soils, but is less effective to change their residual forms. The applications of chelating agents promote the uptakes of Cd and Pb by ryegrass, but reduce its biomass. In general, the results obtained from the pot experiments are reliable. But the English is poorly written with many grammar mistakes, and should be completely revised.

The English expression of the whole manucript should be completely rewritten. I only point out some problems as follows:

Line 28: “the in so”?

It has been corrected in the manuscript.

Line 35-36: grammar mistake.

It has been corrected in the manuscript.

Line 37-38: bad expression.

It has been corrected in the manuscript.

Line 411-444: the conclusions should be shortened.

Response: Thank you very much for your valuable comments and suggestion. We have made corrections and additions to the manuscript carefully. The conclusions also have been deleted.

For the problem of the English expression, we have commissioned an English editing agency to revise the manuscript, the certificate is attached. 

Reviewer 2 Report

The article is understandable and quite interesting to read. Further research is needed, more can be read in the review.

The study provides important insights into the efficacy of chelating agents in enhancing the phytoremediation of Cd and Pb by ryegrass. However, critical concerns about the potential environmental risks associated with the use of chelating agents should be addressed. The study was short in duration and some more plant species could be studied. Further research is needed to thoroughly assess the environmental impact and long-term effectiveness of chelating agents in phytoremediation, as well as their potential impact on ecosystems and human health. A brief explanation on this topic would contribute to the paper.

Corrections needed:

- Chart on row 228 is a missing legend.

 - In rows 59, 60, and 61: Please explain where chelating agents are added.

Author Response

Chart on row 228 is a missing legend.

Response : Thank you so much. We have carefully revised the article and the issues you raised have been added and clarified in the text. The legends have been added to Fig. 2 and Fig. 3 respectively.

- In rows 59, 60, and 61: Please explain where chelating agents are added.

Response: Thanks a million for your comments. The modifications have been made to carry out phytoremediation in which a chelating agent is added to the contaminated soil.

Reviewer 3 Report

Article submitted by Dong et al., entitled “Effects of Chelating Agents Addition on Ryegrass Extraction of Cadmium and Lead in Artificially Contaminated Soil” highlights the removal of cadmium (Cd) and lead (Pb) from the soil by phytoremediation using Ryegrass combined with chelators. After quite investigation, I recommend its publication in this journal (major revision) after providing proper improvement according to the following suggestions, modifications and reply to raised queries.

1.     Please mention the harmful impacts of cadmium (Cd) and lead (Pb) on human and the biosphere.

2.     What are the reasons for choosing these two elements in this research? Explain more in detail.

3.     The introduction requires more improvement; please clarify the motivation, innovation and contribution of this study. More typical studies on phytoremediation using eco-friendly plants for removal of toxic metals are suggested to be cited in introduction with comparison to enrich the various applications of related technology and the subsequent stabilization of such produced toxic bio-waste, e.g.

 10.1080/02757540.2018.1546296, 10.1016/j.pnucene.2020.103285.

Overall, it would be beneficial if the introduction is improved to justify the need for this research and objectives.

4.     Please provide any cost and efficiency comparisons between established removal technique in current study and other traditional approaches to cover the claims.

5.     There is no information about the contact time, give more information about contact time to conduct the optimum time of maximum removal percent.

6.     Extensive article editing is required. There are many things to improve in the text.

7.     It is recommended to extend the comparison of the study findings with other similar published work under the results and discussion section.

8.     By comparing the discussion and the conclusion, it is clearly notified that the discussion is shorter than the discussion!!!! The discussion needs more deep investigation.

Author Response

  1. Please mention the harmful impacts of cadmium (Cd) and lead (Pb) on human and the biosphere.

Response: Thank you very much for your review of the article. We have carefully revised and improved the text in accordance with your comments. The harmful impacts of cadmium (Cd) and lead (Pb) on human and the biosphere have been added in the manuscript.

Cd is one of the most toxic elements to human beings, as it is harmful to bones and kidneys when accumulated over a long period of time. Pb is one of the most abundant toxic metals that exerts the most dangerous effects on all living beings. It has high mobility and is easily taken up by plants even at low contamination levels. The health hazards caused by Cd and Pb pollution take a long time to accumulate before they become apparent, with a time lag. The World Health Organization has classified Cd as a priority food contaminant for research.

  1. What are the reasons for choosing these two elements in this research? Explain more in detail.

Response: Thank you very much. Due to the harmful impacts of Cd and Pb on human and the biosphere, and China also regards Cd and Pb as the key monitoring indicators for implementing total emission control. So, we first choose them as the study elements. In the manuscript, Line 42-45: the total exceeding the standard rate of soil pollution points reached 16.1%, and the over-standard rate of Cd and Pb reached 7.0% and 1.5%, respectively; Cd had the largest proportion among all heavy metal pollutants, and Pb was the fifth among all heavy metals.

3.The introduction requires more improvement; please clarify the motivation, innovation and contribution of this study. More typical studies on phytoremediation using eco-friendly plants for removal of toxic metals are suggested to be cited in introduction with comparison to enrich the various applications of related technology and the subsequent stabilization of such produced toxic bio-waste, e.g.

 10.1080/02757540.2018.1546296, 10.1016/j.pnucene.2020.103285.

Overall, it would be beneficial if the introduction is improved to justify the need for this research and objectives.

Response: Thank you so much. In the light of your suggestions, we have carefully revised the introductory part, and we have read the two papers carefully and cited them.

  1. Please provide any cost and efficiency comparisons between established removal technique in current study and other traditional approaches to cover the claims.

Response: Thank you. We are so sorry that the purpose of this study is just to compare the removal efficiency of five chelating agents on Cd and Pb contaminated soil, in order to select the most suitable chelating agent. Therefore, there is not much consideration given to the remediation cost, and this aspect will be considered for supplementation in future study.

  1. There is no information about the contact time, give more information about contact time to conduct the optimum time of maximum removal percent.

Response: Thank you so much. In the pot experiment of this study, the chelating agents was added to the soil surface of the pots at 45 days of ryegrass growth and the aerial and root parts of the plants were harvested 15 days later. It means that the contact time between the chelating agents and the contaminated soil was 15 days.

  1. Extensive article editing is required. There are many things to improve in the text.

Response: Thank you very much for your valuable suggestions, we have commissioned an English editing agency to revise the manuscript, the certificate is attached.

  1. It is recommended to extend the comparison of the study findings with other similar published work under the results and discussion section.

Response: Thank you very much. We have integrated and supplemented the Discussion and Results parts.

  1. By comparing the discussion and the conclusion, it is clearly notified that the discussion is shorter than the discussion!!!! The discussion needs more deep investigation

Response: Thank you for your reviews. We have integrated and supplemented the Discussion parts, and reduced the Conclusion parts.

Reviewer 4 Report

The article had major experimental design flaws, no control treatment was set up, the English sentences were difficult to understand, the discussion was not insightful and the overall writing was poor.

Bad English writing, needs revision by native speakers

Author Response

The article had major experimental design flaws, no control treatment was set up, the English sentences were difficult to understand, the discussion was not insightful and the overall writing was poor.

Bad English writing, needs revision by native speakers.

Response: Thank you very much for your review. We have carefully revised and improved the text in accordance with your comments, we also have commissioned an English editing agency to revise the manuscript, the certificate is attached.

Round 2

Reviewer 3 Report

 Accept in present form.

Reviewer 4 Report

The author has revised the questions I asked and I agree to accept and publish this manuscript.